# Oxidative Stress Plays an Important Role in Glutamatergic Excitotoxicity-Induced Cochlear Synaptopathy: Implication for Therapeutic Molecules Screening

**DOI:** 10.3390/antiox13020149

**Published:** 2024-01-25

**Authors:** Anissa Rym Saidia, Florence François, François Casas, Ilana Mechaly, Stéphanie Venteo, Joseph T. Veechi, Jérôme Ruel, Jean-Luc Puel, Jing Wang

**Affiliations:** 1Institute for Neurosciences of Montpellier (INM), INSERM U1298, University Montpellier, 34295 Montpellier, France; florence.francois@inserm.fr (F.F.); ilana.mechaly@umontpellier.fr (I.M.); stephanie.venteo@inserm.fr (S.V.); jvecchi@uiowa.edu (J.T.V.); jean-luc.puel@inserm.fr (J.-L.P.); 2INRA, UMR 866 Dynamique Musculaire et Métabolisme, 34060 Montpellier, France; francois.casas@inrae.fr; 3Centre de Recherche en CardioVasculaire et Nutrition, Aix-Marseille Université-INSERM, 1263-INRAE 1260, 13385 Marseille, France; jerome.ruel@univ-amu.fr

**Keywords:** synaptopathy, glutamate excitotoxicity, kainic acid, AMPA receptors, oxidative stress, antioxidants, neurotrophic factor

## Abstract

The disruption of the synaptic connection between the sensory inner hair cells (IHCs) and the auditory nerve fiber terminals of the type I spiral ganglion neurons (SGN) has been observed early in several auditory pathologies (e.g., noise-induced or ototoxic drug-induced or age-related hearing loss). It has been suggested that glutamate excitotoxicity may be an inciting element in the degenerative cascade observed in these pathological cochlear conditions. Moreover, oxidative damage induced by free hydroxyl radicals and nitric oxide may dramatically enhance cochlear damage induced by glutamate excitotoxicity. To investigate the underlying molecular mechanisms involved in cochlear excitotoxicity, we examined the molecular basis responsible for kainic acid (KA, a full agonist of AMPA/KA-preferring glutamate receptors)-induced IHC synapse loss and degeneration of the terminals of the type I spiral ganglion afferent neurons using a cochlear explant culture from P3 mouse pups. Our results demonstrated that disruption of the synaptic connection between IHCs and SGNs induced increased levels of oxidative stress, as well as altered both mitochondrial function and neurotrophin signaling pathways. Additionally, the application of exogenous antioxidants and neurotrophins (NT3, BDNF, and small molecule TrkB agonists) clearly increases synaptogenesis. These results suggest that understanding the molecular pathways involved in cochlear excitotoxicity is of crucial importance for the future clinical trials of drug interventions for auditory synaptopathies.

## 1. Introduction

Spiral ganglion neurons (SGNs) of the cochlea are composed of two types of sensory neurons. Among SGNs, 90–95% are type I afferent neurons, which are myelinated and synapse on inner hair cells (IHCs). Each terminal of type I SGNs innervates only one IHC, while each IHC receives contacts from 10 to 20 terminals of type I SGNs [1]. Type II SGNs represent only 5 to 10% of the afferent neurons [2,3] and are non-myelinated and pseudounipolar [4,5,6]. Each type II SGNs innervates about 15–20 outer hair cells (OHCs) in the same row, while each OHC is contacted by a single type II afferent neuron. IHC synapses are composed of a single postsynaptic density (PSD) juxtaposed to a presynaptic ribbon-type active area in the IHC. PSDs of the IHC synapse are comprised of PSD95 scaffold protein and a few thousand of glutamate receptors receiving excitation from the IHC ribbon synapse [7,8]. SGNs express several types of glutamate receptors, predominantly: α-amino-3-hydroxy-5-methylisoxazole-4-propionic acid (AMPA: GluR2-4), N-methyl- D-aspartate (NMDA: NR1), and kainate (GluR5) [9,10]. Glutamate is identified as the neurotransmitter between IHCs and auditory nerve fibers (ANFs) [11,12,13].

It is widely accepted that glutamate excitotoxicity, induced by excessive activation of AMPA receptors through redundant release of glutamate from overstimulated IHCs, may be sufficient to initiate drastic excitotoxic damage to ANF terminals [14,15,16,17,18]. In support of this hypothesis are findings that local round window membrane application of kainic acid (KA), a full glutamatergic agonist, also induces severe swelling of auditory terminals below the IHCs [14,19,20]. Exposure to noise causes disrupted expression of vesicle transporter protein 3 (Vglut3), which is known to play a crucial role in the release of glutamate from inner hair cells into the synaptic cleft [21]. Noise also causes a disruption in the expression of the glutamate/aspartate transport protein (GLAST), responsible for glutamate absorption, and of the Na/K-ATPase a1 coupled to GLAST, during synaptopathy in the cochlea [21]. Moreover, genetic deletion of GLAST in mice resulted in greater synaptic swelling after noise overexposure [22,23]. By contrast, the non-NMDA ionotropic glutamate receptor antagonist, kynurenate, reduces noise damage to synapses [14], and genetic deletion of Vglut3 prevents synapse loss after noise exposure in mice [24]. Taken together, these results suggest that glutamate excitotoxicity is a major factor in the damage of IHC synapses [25,26,27].

Synaptopathy is generally thought to be an early pathological finding in noise-induced, ototoxic drug-induced, or age-related hearing loss. Synaptopathy is implicated in the impaired perception of auditory signals in the presence background noise in mice and humans [28,29,30,31,32,33,34,35,36,37,38,39,40]. While it has been demonstrated that AMPA receptor over-activation is sufficient to cause synaptopathy, the excitotoxic mechanisms of glutamate in the cochlea are not completely understood. How excessive synaptic glutamate release leads to noise- or ototoxic drug-induced or age-related synapse loss is not fully understood. The requirement of endogenous glutamate for synapse loss remains unclear. The understanding of signaling pathways involved in excitotoxicity-induced cochlear synaptopathy is crucial in the search for tools to prevent or treat cochlear synaptopathy.

Here, our working hypothesis is that overstimulation of IHCs by noise may lead to massive calcium influx, which can induce activation of the calpain pathway and cause mitochondrial damage and subsequent increase in levels of oxidative stress and pro-inflammatory process. Together, this leads to neurodegeneration of the auditory nerve terminals and synaptopathy (Figure 1). To test this hypothesis, we examined the molecular basis responsible for KA-induced loss of IHC synapses and degeneration of the auditory nerve terminals of type I SGNs using a cochlear explant culture from P3 mouse pups. In addition, we assessed the efficacy of antioxidant and neurotrophic factors, such as NT3, BDNF, and some mimetics of BDNF, on cochlear synapse regeneration.

## 2. Materials and Methods

### 2.1. Animals

A total of 100 pregnant female Swiss mice were purchased from Janvier Laboratories (Le Genest-Saint-Isle, France) and were housed in pathogen-free animal care facilities accredited by the French Ministry of Agriculture and Food (D-34-172-36, 20 May 2021). Each pregnant mouse gives birth to a litter of around ten babies on average. Experiments were carried out on third-day neonatal mouse pups (P3) without distinguishing their sex. All protocols comply with French Ethical Committee stipulations regarding the care and use of animals for experimental procedures (agreements C75-05-18 and 01476.02, license #6711). All efforts were made to minimize the number of animals used.

### 2.2. Organ of Corti, and Whole Cochlea Cultures

Mouse whole cochleae and the organ of Corti explants were collected from P3 mice and prepared according to the procedures described previously [41,42]. The whole cochleae were kept in suspension, and the organ of Corti explants were kept in adherent conditions in a 6-well culture plate containing 2 mL/well of culture medium. The culture medium consisted of Dulbecco’s modified Eagle’s medium/nutrient mixture F-12 (DMEM/F-12, #21331020) containing 2 mM L-glutamine (#25030024), N-2 complement (#17502048) at 1X and insulin transferrin selenium (#41400045) at 1X purchased from Gibco Life technologies (Thermo Fisher Scientific group, Illkirch, France), and 8.25 mM D-glucose (#G6152) and 30 U/mL penicillin G (#P3032) from Sigma-Aldrich (Merck group, Saint Quentin Fallavier, France).

### 2.3. KA-Induced Excitotoxic Damage

KA was purchased from Tocris Bioscience (#0222, Bio-Techne group, Bristol, Royaume-Uni) and was freshly prepared in culture medium to final concentrations of 0.5 mM. To cause excitotoxic trauma, after 24 h in culture, the organ of Corti explants were exposed to 0.5 mM KA for 2 h. This concentration was extrapolated from previous in vitro studies [43,44]. To determine time-dependent IHC synapse loss, after 2 h KA treatment, organ of Corti explants were then maintained in the culture medium alone for 0, 12, 24, 48, and 72 h. All control samples were incubated in culture medium alone and were performed simultaneously with the experimental cultures. The cultures were then fixed and prepared for immunofluorescence imaging.

### 2.4. Pharmacological Drug Preparation and Interventions

Resveratrol, brain-derived neurotrophic factor (BDNF), neurotrophin 3 (NT3), and tropomyosin receptor kinase B (TrkB) agonist LM22A-4 were purchased from Sigma-Aldrich (#R5010), (#SRP6037), (#N1905) and Tocris (#4607), respectively. R13, a TrkB ligand prodrug, was purchased from Axon Medchem (#3775, Groningen, Netherlands). CF3CN, an optimized synthetic 7,8-DHF chemical, which has been reported as a pharmaceutical candidate for treating Alzheimer’s disease [45,46], was provided by AB Science. Resveratrol was prepared in acetone at 10 mM; BDNF and NT3 at 10 µg/mL and 5 µg/mL in PBS 1X with 1% BSA, respectively; and LM22A-4, R13, and CF3CN in water at 10 mM. These molecules were then freshly diluted in culture medium at final concentrations ranging from 0–10–20 µM and 0–1–5–10 µM for resveratrol and the TrkB agonists (LM22A-4, R13, and CF3CN), respectively. The final concentrations of resveratrol and TrkB agonists were chosen based on our preliminary evaluations of the dose–response effects of these molecules against KA-induced IHC synapse loss on day 2 in KA-exposed cochlear explants. The concentrations of BDNF and NT3 were chosen based on previous published in vitro reports [44].

For pharmacological interventions, after KA treatment, the explants were maintained up to 48 h in control culture medium or in culture medium containing BDNF (10 nM), NT-3 (10 nM), resveratrol (10 or 20 µM), LM22A-4 (1, 5 or 10 µM), CF3CN (1, 5 or 10 µM), or R13 (1, 5 or 10 µM), respectively. All control samples were maintained in culture medium alone and were run concurrently alongside the experimental cultures. The cultures were then fixed and prepared for immunofluorescence.

### 2.5. Counting of Ribbon Synapses of IHCs

Immunocytochemistry was performed in cultured treated and untreated control organs of Corti using antibodies (Appendix A) against CtBP2 (1:500, BD Biosciences, Grenoble, France, #612044 RRID:AB_399431) and PSD95 (1:500, NeuroMab, UC Davis/NIH NeuroMab Facility, Ca, USA, #75-028 RRID:AB_2292909) to identify the presynaptic ribbons and postsynaptic density, respectively. The samples were counterstained with anti-VGLUT3 (1:500, Synaptic Systems, Göttingen, Allemagne, #135204 RRID:AB_2619825) to label the inner hair cells. All secondary antibodies were used at a dilution of 1:1000. These included goat anti-mouse IgG1 and IgG2a and goat anti-guinea pig IgG conjugated to Alexa 488, Alexa 568, or Alexa 647, respectively (Molecular Probes, Thermo Fisher Scientific group, Illkirch, France, #A21121 RRID:AB_2535764, #A21134: RRID:AB_2535773, # A21450 RRID:AB_2535867). DNA was stained using Hoechst 33342 (1:5000, Thermo Fisher Scientific group, #62249). To exclude nonspecific binding of the secondary antibodies, negative controls were performed without primary antibodies. Fluorescent tags were visualized using confocal microscopy (Zeiss 880 Airyscan, Zeiss group, Germany) with a Plan-Apochromat 63X/1,4 Oil DIC M27 objective and imaged with a 4-channel z-stack spanning the height of the hair cells to capture all synaptic puncta. The confocal images were acquired with 1024 pixels × 1024 pixels in x and y, with z-spacing at 0.32 µm per slice (Appendix A). The quantitative analyses (6 to 10 cochleae per condition and per time point) were performed in over 10 successive IHCs in the cochlear regions centered at 1.1, 2.6, or 4.1 mm from the cochlear apical end and corresponding to frequencies of 8, 16, and 32 kHz, respectively [47]. The quantitative analysis of ribbons and synapses, i.e., juxtaposed spots of the presynaptic ribbon component RIBEYE and postsynaptic density protein PSD95, were carried out in z-stacks from confocal imaging using a custom 3D algorithm [48].

### 2.6. Counting of Terminals of Auditory Nerve Fibers

Immunocytochemistry was performed in cultured organs of Corti using antibodies (Appendix A) against anti-VGLUT3 (1:500, Synaptic Systems, Göttingen, Allemagne, #135204 RRID:AB_2619825) to label IHCs and anti-NF200 (1:400, Sigma-Aldrich, Merck group, Saint Quentin Fallavier, France, #N0142 RRID:AB_477257) to label the SGNs and their peripheral axons. All secondary antibodies were used at a dilution of 1:1000. These included goat anti-guinea pig and donkey anti-rabbit IgG conjugated to Alexa 488 and Alexa 594, respectively (Molecular Probes, Illkirch, France, #A-11073 RRID: AB_2534117, #A-21207, RRID:AB_141637). DNA was stained using Hoechst 33342 (1:5000, Thermo Fisher Scientific group Illkirch, France, #62249). Fluorescent tags were visualized using confocal microscopy (Zeiss 880 Airyscan). The quantitative analyses of NF200-labeled fibers (6 cochleae per condition and per time point) were carried out in the cochlear regions coding 32 kHz, each containing 10–15 hair cells. The terminals are quantified as the average number of NF200-labeled fibers in contact with an IHC.

### 2.7. Measurement of Enzymatic Activities and Oxidative Stress

Enzymatic activities and oxidative stress were assessed in whole cochlea homogenates as previously described [42,49]. Catalase activity was measured as previously described by Marklund [50] and was expressed in U/mg protein. Citrate synthase and Cytochrome oxidase activities were evaluated as previously described and were expressed in mU/mg protein [51,52]. Lipid peroxidation was evaluated using the thiobarbituric acid-reactive substances method and was expressed in mol/mg malondialdehyde (MDA) [53]. The concentration of Thiol (SH-group) levels was measured as previously described [54] and was expressed in nmol/mg protein. Each experiment consisted of a pool of 6 cochleae from 3 mice per sample and were performed in 8 biological replicates.

### 2.8. Immunocytochemistry

Immunocytochemistry was employed to probe the abundance of cytochrome c oxidase and catalase in control and KA-treated cochlear explants, as well as the cellular localization of BDNF and TrkB in cryostat sections of untreated p3 mouse cochleae. The primary antibodies (Appendix A) used were anti-cytochrome c oxidase (1:200, BD Biosciences, #556432 RRID:AB_396416), anti-BDNF (1:500 Abcam, Amsterdam, Netherlands, #ab223354), and anti-TrkB (1:500, Abcam, #ab18987 RRID:AB_444716). The samples were counterstained with Alexa 647 phalloidin (1:1000, Thermo Fisher Scientific, #A22287) to visualize filamentous actin and Hoechst 33342 (1:5000, Thermo Fisher Scientific group Illkirch, France, #62249) to stain the nuclei. The secondary antibodies used were as follows: donkey anti-mouse and anti-rabbit IgG conjugated to Alexa 488 (Molecular Probes, Illkirch, France, #A-21202 RRID: AB-141607). Fluorescent tags were visualized using confocal microscopy (Zeiss 880 Airyscan, Zeiss group, Germany). In control samples without primary antibodies, Alexa 488 fluorescent tags were not observed. Immunocytochemistry assessments required 3 to 5 cochleae per condition.

### 2.9. Immunoblotting

The lysate of the cochlear samples was obtained using RIPA buffer (Thermofisher #89900). The protein concentration was evaluated by Pierce™ BCA Protein Assay Kits (Thermo Fisher group Illkirch, France, #23225). Cochlear homogenates were prepared in 4X Laemmli sample buffer (Biorad, Marnes-la-Coquette, France, #1610747) for a ratio of ¼ protein/Laemmli sample buffer. The protein was then denatured for 5 min at 95 °C. A weight of 15 µg of protein was loaded per lane in a pre-made gel (Biorad, Any kD MP TGX Stain-Free 10W 50 μL pkg 10 #4568124). The electrophoresis apparatus (Biorad) was filled with 1X Tris/glycine/SDS running buffer to separate protein samples by SDS-PAGE (Biorad, #1610772). The gel was run at 90 V for 10 min, and then 120 V for 50 min. The protein transfer was performed using PVDF membrane (Biorad, Trans-Blot Turbo Midi PVDF Transfer Packs #1704157) in Biorad apparatus (Trans-Blot^®^ Turbo™ Transfer System #1704150) according to the manufacturer’s instructions.

The membranes were rinsed and then incubated in blocking solution (milk 5% in 1X TBST) for 1 h at room temperature. Next, they were incubated with primary antibodies overnight at 4 °C in 1X TBST with 5% milk. The membrane was washed three times with TBST (purchased from Fisher Bio regeant #BP2471-1, TWEEN^®^ 20 from Sigma-Aldrich #P5927). The primary antibodies used were: antibodies recognizing 4-hydroxynonenal (4HNE, 1:1000, Bioss Antibodies, Woburn, MA, USA, #bs-6313R RRID:AB_2827741), Ras homolog gene family member A (RhoA, 1:1000, Cell Signaling, Danvers, MA, United States, #2117 RRID:AB_10693922), phospholipase C-γ (PLC-γ, 1:1000, Cell Signaling, #2822 RRID: AB_2163702), AKT (1:1000, Cell Signaling, #4691 RRID:AB_915783), and pAKT (1:1000, Cell Signaling, #4060 RRID:AB_2315049). β-actin (1:10000, Sigma-Aldrich, Merck group, Saint Quentin Fallavier, France, #A1978 RRID: AB-476692) served as a loading control.

The membranes were then incubated with the secondary antibodies for 2 h at room temperature. The secondary antibodies used were horseradish peroxidase-conjugated goat anti-mouse IgG (1:3000, Jackson Immuno Research, Europe group, Cambridge, United Kingdom, #115-001-003 RRID: AB-2338443), or goat anti-rabbit IgG (1:3000, Jackson ImmunoResearch, #111-001-003 RRID: AB-2337910). Finally, the membranes were washed three times with TBST. The detection of protein was performed using a chemiluminescent reagent (Thermo Fisher scientific group, Illkirch, France, SuperSignal™ West Pico PLUS Chemiluminescent Substrate, #34580) and Vilber fusion Fx device.

Image scans of the Western blots were used for semi-quantitative analysis with Fiji software (Version 2.14.0/1.54f). Each experiment consisted of a pool of 10 cochleae from 5 mice per sample and was performed in a biological triplicate and 2–3 technical replicates. All results were normalized by β-actin expression.

### 2.10. Quantitative PCR

Total RNA was extracted from 10 cochleae per sample using a TRI-reagent (Invitrogen, Thermo Fisher Scientific group, Illkirch, France, # 9738G) according to the manufacturer’s instruction. Concentration and purity were assessed using a NanoDrop spectrophotometer (Thermo Fisher Scientific, Rockford IL, USA) with ratios of A260/A280 and A260/A230. Samples were reverse-transcribed using a PrimeScrip RT Reagent Kit (Qiagen, #330401). Real-time PCR was carried out using SYBR Green I dye detection on a Light Cycler system (Roche Molecular Biochemicals, Meylan, France). PCR reactions were carried out in 96-well plates in a 10 µL volume containing 3 µL of cDNA product (final dilution 1/30), 0.5 µM of forward and reverse primers, and 2 µL of QuantiTect SYBR Green PCR Master Mix (Roche Diagnosis, Meylan, France). The sequences of the primer pairs used are listed in Table 1. All experiments were performed in technical triplicate on two independent RT reaction products. The relative amounts of specifically amplified cDNAs were calculated on at least three independent experimental replicates using the delta-CT method [55] and were normalized with polymerase (RNA) II polypeptide J, Polr2j, and DEAD box polypeptide 48, Ddx48, as stable control genes. In particular, arbitrary units (a.u.) of each gene expression were calculated through normalization with a factor. This normalization factor represents the geometric averaging of the expression of two stable control genes as described by Vandesompele et al. [55].

## 3. Statistics

Data are expressed as the mean ± SEM. The Shapiro–Wilk test was used for normalization of the variables. The significance of the group differences was evaluated with a one-way ANOVA; once the significance of the group differences (*p* ≤ 0.05) was established, Dunn’s tests were used for post hoc comparisons between pairs of groups. Based on data from our previous reports [42] or from preliminary experiments, we calculated the sample size using G*Power 3.1.9.2 to ensure adequate power of key experiments for detecting pre-specified effect sizes.

## 4. Results

To properly study the excitotoxic mechanisms of glutamate in the cochlea and to develop effective therapies for synapse repair or regeneration, we adopted an available in vitro model of excitotoxicity [43]. This model consists of exposing cochlear explants or whole cochleae to KA, a full agonist of AMPA/KA-preferring glutamate receptors, to induce the loss of IHC synapses and degeneration of the auditory terminals of type I SGNs in the cochlear explants or whole cochleae in culture from P3 mouse pups.

### 4.1. Short-Term KA Exposure Induces Rapid IHC Synaptopathy

To determine the contribution of short-term KA exposure on the loss of ribbon synapses, we examined the cultured KA-exposed and unexposed cochlear explants by triple-labeling presynaptic and postsynaptic structures and IHCs, as well as 3D confocal imaging analysis [48] in the cochlear regions coding 8, 16, and 32 kHz frequencies.

Our results showed that control, unexposed explants displayed 23.66 ± 2.24, 31.24 ± 1.61, and 38.61 ± 0.74 ribbons/IHC, and 20.18 ± 1.41, 26.31 ± 0.90, and 24.18 ± 1.43 paired synaptic puncta (synapses) per IHC at cochlear regions coding 8, 16, and 32 kHz frequencies, respectively (Figure 2A–G, Appendix A, and Figure 3D–F) after two hours (+1 day of rest) in culture. Two-hour-KA exposure caused a significant loss of synapses in the cochlear regions coding 16 kHz (18.9 ± 1.9 synapses/IHC, *p* ≤ 0.01, n = nine cochleae from three different experiments) and 32 kHz (18.2 ± 1.5 synapses/IHC, *p* ≤ 0.05, n = nine cochleae from three different experiments), but not 8 kHz (*p* > 0.05, n = nine cochleae from three different experiments, Appendix A) compared with the control explants (Figure 2D–G and Appendix A). In contrast, a similar number of ribbons/IHC was observed in both the control and KA-exposed cochlear explant after two hours of treatment (Figure 3A–C). Furthermore, IHCs, OHCs, and SGNs seem intact up to 48 h in culture. These results are consistent with previous in vivo and in vitro studies [43,56]. The addition of NMDA, another glutamate receptor agonist (0.5 mM), in the culture medium did not affect the action of KA on IHC synapses in the regions coding 32 kHz and even slightly, but not significantly, improved synapse survival in 16 kHz coding region in two-hour-KA exposed cochlear explants (Appendix A). Together with our previous in vivo study [18,56], our results suggest that NMDA receptors are clearly not involved in cochlear excitotoxicity. In the following experiments, we, therefore, evaluated the effects of KA alone.

### 4.2. Time Course of KA-Induced IHC Synapse Loss

The counting of remaining synapses per IHC in control explants revealed a time-dependent synapse loss, which reached significance at 72 h in culture. Two-hour-KA exposure induced a significantly greater loss of synapses immediately after exposure and this decrease was maintained up to 72 h after exposure (2 h: 16 kHz: *p* ≤ 0.01, 32 kHz: *p* ≤ 0.05; 12 h: 16 kHz: *p* ≤ 0.0001, 32 kHz: *p* ≤ 0.001; 24 h: 16 kHz: *p* ≤ 0.001, 32 kHz: *p* ≤ 0.0001; 48 h: 16 kHz: *p* ≤ 0.001, 32 kHz: *p* ≤ 0.01; and 72 h: 16 kHz: *p* ≤ 0.05, 32 kHz: *p* ≤ 0.001, n = 9 cochleae from 3 independent experiments, Figure 2D,F). Additionally, when compared with immediately after (2 h), a significantly greater loss of synapses was seen from 48 h and 24 h for the 16 and 32 kHz regions, respectively, in KA-exposed explants (Figure 2D,F). Approximately, just greater than a 50% synapse loss was observed in the regions coding 16 and 32 kHz in KA-exposed cochlear explants compared with control explants for all post-exposure-times (Figure 2E,G).

A progressive and significant reduction in the number of ribbons/IHC was seen in both KA-treated and untreated cochlear explants after 48 h in culture compared to the numbers immediately after two-hour KA exposure or two hours in culture in both cochlear regions coding 16 and 32 kHz (16 kHz: KA 48 h vs. KA 2 h or Ctrl 48 h vs. Crtl 2 h: *p* ≤ 0.001; 32 kHz: KA 48 h vs. KA 2 h: *p* ≤ 0.001; Ctrl 48 h vs. Crtl 2 h: *p* ≤ 0.0001, n = nine cochleae from three independent experiments) (Figure 3A–C). We also observed a significant reduction in NF200-labelled type I auditory nerve fiber terminals from KA-treated cochlear explants compared to control explants 72 h after exposure (*p* ≤ 0.001, n = six cochleae from three independent experiments, Figure 3D–F). 

### 4.3. KA Causes Oxidative Stress

Mitochondria play a key role in cochlear homeostasis and in maintaining cell function during exposure to auditory stress, such as noise trauma. Here, the effects of KA-exposure on mitochondrial activity were measured through the assessments of complex IV of the mitochondrial respiratory chain (Mt CxIV, or cytochrome c oxidase), catalase (a first-line antioxidant enzyme), and citrate synthase (CS, a marker of mitochondrial density) via comparing their activities in KA-exposed and unexposed cochleae. Our results showed that KA exposure caused a significant increase in mt CxIV and catalase activities at 48 h after exposure compared with control unexposed and cultured cochleae (KA + 48 h vs. Crtl: *p* ≤ 0.01; KA + 48 h vs. Crtl + 48 h: *p* ≤ 0.05, n = eight biological replicates, Figure 4A,B). On the other hand, no significant difference in the level of CS activity was found between groups (Figure 4C). The increase in the activity of mt CxIV was consistent with confocal microscopy observations showing an intense expression of cytochrome c oxidase mainly in the cytoplasm of the SGNs 48 h after KA exposure (Figure 4D,E).

To investigate the occurrence of oxidative stress in cochlear tissues after KA exposure, lipid peroxidation and protein oxidation were evaluated by measuring the levels of malondialdehyde (MDA) and thiols (SH). We observed that MDA levels were significantly increased in KA-exposed cochleae 48 h after exposure compared with control unexposed cochleae (*p* ≤ 0.01, n = eight biological replicates, Figure 4F), while no significant difference in SH levels was seen between KA-exposed and unexposed cochleae (Figure 4H). These results suggest that KA induced lipid peroxidation. Western blot analyses demonstrated a significant increase (*p* ≤ 0.05, n = five independent experiments including biological triplicates and one to two technical replicates) in the level of 4HNE, one of the lipid peroxidation products, 48 h after exposure compared with control unexposed cochleae (Figure 4G and insert). Altogether, these results suggest that events linked to oxidative stress occur in the cochlea after exposure to KA.

### 4.4. Antioxidant Treatment Prevent KA-Induced Acute Synaptopathy and Promote Synapse Regeneration

We also considered the role of this work in developing pharmacological therapies to prevent excitotoxicity-induced cochlear synaptopathy or to promote synapse regeneration. We, therefore, investigated the potential therapeutic effects of resveratrol, one of the most highly investigated antioxidant molecules [57]. Interestingly, co-treatment of the cochlear explants with 0.5 mM KA plus 10 µM resveratrol completely prevented KA-induced and 48-hour culture condition-induced synapse loss. In particular, the number of synapses per IHC in the 16 and 32 kHz coding regions from co-treated explants 48 h after exposure was greater than the control conditions in the same regions and at the same time point. Remarkably, these synapse counts were similar to the control 2 h in culture (Figure 4I–K). Finally, the treatment of explants immediately after two-hour-KA exposure with resveratrol at concentrations ranging from 10 to 20 µM also significantly (*p* ≤ 0.0001, n = 8 cochleae from 3 independent experiments) rescued KA-induced IHC synapse compared with KA alone (Figure 4I–K). Resveratrol treatment even rescued the synapse loss induced by the culture condition to reach a similar number of synapses in the control 2 h (Figure 4J,K). Together, these results strongly suggest that counteracting free radical damage may be a promising approach to prevent or treat cochlear excitotoxicity-induced IHC synaptopathy.

### 4.5. KA Causes Reduced Expression of BDNF, NT3, and Their Trk Receptors, While Increasing Expression of NGF

It has long been known that neurotrophic growth factors like BDNF and NT3 are essential for the survival of the SGN and the guidance of their fibers to the sensory epithelia of the embryonic cochlea [58,59]. However, much less is known about the function of these neurotrophins in the postnatal cochlea, particularly in the excitotoxic damaged cochlea. As already observed in a previous publication [60], we showed that BDNF was mainly expressed in IHCs and OHCs to some extent in the spiral ganglion cells of postnatal P3 mouse cochlea (Figure 5A). Similarly, a higher expression of its receptor tropomyosin type B-related kinase (TrkB) was also mainly distributed in IHCs with a lower level in OHCs and spiral ganglion cells (Figure 5B).

To examine if there was any effect of KA treatment on the expression of neurotrophic factors and their receptors, we compared the abundance of their transcripts in KA-treated and untreated cochleae. Our results showed a significant reduction in the expression of *BDNF* immediately after two-hour KA exposure and this was maintained to 48 h after (KA vs. Crtl: *p* ≤ 0.0001; KA + 48 h vs. Ctrl: *p* ≤ 0.001, vs. Crtl + 48 h: *p* ≤ 0.05, n = six qPCR from two RT reactions). A decrease in the expression of TrkB was only observed immediately after KA exposure (Figure 5C,D). A significant reduction (*p* ≤ 0.0001, n = six qPCR from two RT reactions) of NT3 expression was observed in control unexposed cochleae 48 h in culture (Figure 5E). A significantly lower level of NT3 was noted immediately after KA exposure compared with control explants at the same time point (2 h in culture). Although, a significant recovery of NT3 expression was seen 48 h after KA exposure compared with control cochleae 48 h in culture (*p* ≤ 0.001, n = six qPCR from two RT reactions); at this time point, the NT3 level was still lower than control at 2 h in culture (*p* ≤ 0.05, n = six qPCR from two RT reactions, Figure 5E). Concerning TrkC expression, after 48 h in culture, the cochleae exposed or unexposed to KA displayed a significant reduction of TrkC expression compared to the control cochleae (*p* ≤ 0.0001, n = six qPCR from two RT reactions, Figure 5F).

A previous study reported that NGF may be involved in noise-induced inflammatory responses [61]. In line with this, here we showed that KA exposure induced a time-dependent significant increase in NGF expression (KA vs. Ctrl: *p* ≤ 0.05; KA + 48 h vs. Ctrl, or Ctrl + 48 h: *p* ≤ 0.0001, n = six qPCR from two RT reactions, Figure 5G), but significantly reduced the expression of its TrkA receptor (KA vs. Ctrl: *p* ≤ 0.0001; KA + 48 h vs. Ctrl: *p* ≤ 0.0001, n = six qPCR from two RT reactions, Figure 5H). Forty-eight hours in culture also caused a significantly reduced level of TrkA (*p* ≤ 0.0001, n = six qPCR from two RT reactions, Figure 5H). Together, these results suggest these trophic factors are likely implicated in the cellular responses to the excitotoxic insult, as well culture condition.

### 4.6. Activation of the Downstream Pathways of Trk

It has been proposed that RhoA, a mediator of neuronal death, may be involved in excitotoxic neuronal death subsequent to a reduced level of BDNF in cultured hippocampal and striatal neurons [62]. Here, we revealed that KA exposure caused RhoA protein levels to significantly increase 48 h after exposure compared to control conditions (*p* ≤ 0.01, n = six independent WB experiments including biological triplicates and two technical replicates, Figure 6A,B). Furthermore, significant increases in PLCγ levels and in AKT phosphorylation at Ser463 were also observed in KA-treated cochleae at 48 h after exposure compared to control conditions (*p* ≤ 0.05, n = nine independent WB experiments including biological and technical triplicates, Figure 6A,C,D). PLCγ and AKT are two major molecules in the canonical downstream signaling pathways of Trk signaling pathways and are involved in cell survival [63]. Collectively, our results suggest that KA exposure induces a rapid reduction in expression of *BDNF*, *NT3*, and of their receptors, whilst conversely, an increasing expression of *NGF* and, subsequently, the activation of downstream pathways involved in neuronal death and survival.

### 4.7. Neurotrophic Factor Treatment

We, therefore, investigated the efficacy of exogenous BDNF and NT3 to rescue the IHC synapses from KA-induced excitotoxicity. Our results showed that treating explants immediately after two-hour KA exposure with NT3 and BDNF at 10 nM significantly (*p* ≤ 0.0001, n = 10 cochleae from three independent experiments) rescued KA-induced IHC synapse loss in the 16 kHz coding region at 48 h after exposure compared with KA alone (Figure 7A,B). BDNF, but not NT3, also significantly and completely saves synapses from KA-induced loss in the 32 kHz coding region 48 h after exposure (Figure 7A,C). Based on these results, we thus investigated small molecule agonists of TrkB such as LM22A-4, a TrkB agonist; R13, a TrkB ligand prodrug; and CF3CN, an optimized synthetic 7,8-DHF chemical, to improve synapse regeneration. These molecules have been tested at concentrations ranging from 1 to 10 µM. Our results demonstrated that these small molecule TrkB agonists effectively promoted the regeneration of synapses in the 16 and 32 kHz regions 48 h after exposure to KA with an effective concentration of 1 µM for LM22A-4 and CF3CN and 5 µM for R13 (Figure 7D–I). A comparison of the therapeutic efficacy of BDNF, NT3, and small molecule TrkB agonists confirmed a trend of greater therapeutic efficacy of BDNF and CF3CN compared to other molecules tested in promoting synapse regeneration in the 16 and 32 kHz regions 48 h after KA exposure (Figure 7J,K). Interestingly, our molecular evaluation showed that the administration of BDNF, NT3, and small molecule TrkB agonists significantly reduced the KA-caused increase in the activities of mt CxIV and catalase, as well as the levels of lipid peroxidation 48 h after KA exposure to reach the levels of the control condition (*p* ≤ 0.01, n = 8 replicates, Figure 8A–C).

## 5. Discussion

The IHC synapses are thought to be among the most vulnerable components to both noise-induced and age-related hearing loss [29,36,64]. After synaptic noise or ototoxin exposure, the loss of synapses is immediate and irreversible and can reach more than 50% of the IHC synapses without affecting the sensory hair cells and the hearing thresholds [29,48]. In contrast, the death of the cell bodies of SGNs occurs over years in mice [29]. These observations suggest that there is a therapeutic window through which hearing disorders associated with synaptopathies (hyperacusis, tinnitus, difficulty understanding in noise, and early onset of age-related hearing loss) could be addressed with therapeutic molecules specifically designed to target the key molecular pathways mediating the cochlear synaptopathy.

### 5.1. KA-Induced-Excitotoxicity Leads to Rapid and Irreversible Loss of IHC Synapses

Using an adapted in vitro model of glutamatergic excitotoxicity [43], we showed that two-hour KA exposure caused significant loss of synapses in the cochlear regions coding 16 kHz and 32 kHz, but not 8 kHz. In addition, a loss of the auditory nerve fiber terminals was also observed at 72 h after KA exposure. These results are consistent with previous results obtained from glutamatergic excitotoxic trauma in in vitro and in vivo models [43,65,66]. Here, we demonstrated that co-treatment of the cochlear explants with NMDA and KA did not modify the KA-induced excitotoxic damage to the peripheral processes of SGNs. These results confirmed that excitotoxic damage to the peripheral processes of type I SGNs was mainly mediated by non-NMDA glutamate receptors as previous reports proposed [14,19,56,66]. In addition to the acute loss of synapses induced by 2 h KA exposure, we revealed a slow and progressive loss of synapses as culture time increased in both KA-exposed and unexposed cochleae. In both conditions, no recovery of synapses can be observed for up to 72 h in culture. These results are in agreement with the in vivo results from noise-damaged mouse cochleae [29], but not noise-exposed guinea pig [14] or cat cochleae [67], nor KA-damaged rat cochleae, in culture [43]. The most suitable explanation for this discrepancy between mouse in vitro (our present study) and in vivo [29] studies and early in vivo noise-damaged guinea pig and cat cochleae and KA-damaged postnatal rat cochleae in culture could be due to the species difference. In addition, we showed that KA exposure did not affect the synapse survival in the apical part of the cochlea (coding 2 to 8 kHz). If this is the case for glutamatergic excitotoxic trauma in vivo in guinea pig [65,66], one would expect that it would be difficult to detect synaptic loss in not-obviously-damaged cochlear regions via ultrastructural studies of random transverse sections.

### 5.2. Excitotoxicity-Induced Oxidative Stress and Antioxidant Treatment

Growing evidence links oxidative stress to age-related and noise-induced cochlear sensorineural cell damage and hearing loss [42,68,69,70]. Noise exposure-induced immediate occurrence of O_2_^−^, nitrotyrosine (NT), 4-HNE, and/or OH may persist for several days, leading to a widening of the area of morphological damage [71,72]. Noise exposure also induced increased levels of the oxidative DNA damage marker, 8-hydroxy-2′-deoxyguanosine (8OHdG), and the lipid peroxidation product, 8-isoprostane, in cochlear tissues [73,74]. Consistent with these reports, we revealed that short-term KA exposure led to a significant increase in the activities of mitochondrial complex IV and catalase, as well as the level of MDA and 4HNE 48 h after exposure. Even though the mechanisms of excitotoxicity-induced cochlear synaptopathy have not been exhaustively delineated, based on our results and previous evidence, we can form the hypothesis that increased ROS production may play an important role. In line with this, the addition of glutamate in rat cerebellar granule cells [75,76] and in human SH-SY5Y neuroblastoma cells following the depletion of intracellular glutathione (GSH) levels [77] also increases ROS production. More importantly, our results demonstrate that co-treatment or post-treatment with resveratrol, an antioxidant-like compound, prevents and rescues the KA- and culture condition-induced loss of synapses. Together, these results strongly support our hypothesis that the glutamatergic excitotoxicity-induced loss of synapses is mediated, at least in part, by an increased production of ROS and oxidative stress and that cochlear synaptopathy could be mitigated by antioxidant treatments.

### 5.3. Elicitation of Neurotrophic Effects for the Treatment of the Cochlear Synaptopathy

A number of studies have demonstrated that the altered expression and action of neurotrophic factors and their receptors, such as BDNF/TrkB signaling, is involved in the pathogenesis of neurodegenerative diseases [78,79]. Downregulation of BDNF transcription was observed in human neuroblastoma cells after exposure to oligomeric amyloid beta, one of the risk factors in the pathogenesis of Alzheimer’s disease [80]. Furthermore, BDNF and TrkB mRNA and/or protein levels are markedly reduced in the cortex, striatum, and hippocampus of Huntington’s disease patients and mouse models [63]. Here, we discovered that KA exposure reduced the levels of the mRNA of BDNF, NT3, and their Trk receptors, while increasing the levels of NGF mRNA in the cochleae. KA treatment also led to changes in the amount of proteins or activation of the downstream key elements of Trk signaling pathways like RhoA, PLCγ, and pAKT [63]. These results indicate that the trophic factors are involved in the cellular responses of the cochlea to the KA-induced excitotoxic insult to activate the downstream pathways involving in neuronal death, plasticity, and survival.

As expected, both NT3 and BDNF displayed beneficial therapeutic effects on repair and/or regeneration of damaged afferent fibers of SGNs and IHC synapses, as already shown in noise- or ototoxin-induced hearing loss in vivo [81,82,83,84,85,86]. Since BDNF showed greater synapse regeneration in our experimental paradigm, we, thus, evaluated the therapeutic effects of several novel mimetics of BDNF. Our results provide conclusive evidence that the application of R13, LM22A-4, or CF3CN induced neural outgrowths to completely reestablish the synaptic connection between the IHCs and the auditory terminals of type I SGNs. Finally, the exogenous administration of BDNF, NT3, and BDNF mimetics reduced oxidative stress in cochlear cells. Thus, the neuroprotective actions of BDNF and NT3 in cochlear neural tissues may be mediated by a complex cascade of Trk signaling pathways related to damage repair and neurite regeneration.

## 6. Conclusions

In the present study, we showed that short-term exposure of the postnatal mouse cochlea to kainic acid induced rapid excitotoxicity and the loss of IHC synapses, and later resulted in the degeneration of the distal peripheral axons of type I SGN peripheral axons. KA-induced disruption of the synaptic connection between IHCs and SGNs is associated with an increase in the levels of oxidative stress and alteration of several key elements involved in neurotrophin signaling pathways. Importantly, co-treatment or post-treatment with resveratrol, an antioxidant-like compound, prevents and rescues the KA-induced loss of synapses. This suggests that glutamatergic excitotoxicity may be mediated by an increased production of ROS and oxidative stress and antioxidant treatments could mitigate this cochlear synaptopathy. Finally, the exogenous administration of BDNF, NT3, and BDNF mimetics reduced oxidative stress in cochlear cells and enhanced regeneration of the synapses. Together, these results suggest that understanding the molecular pathways involved in cochlear excitotoxicity is of crucial importance for future clinical trials of drug interventions for auditory synaptopathies.

## Figures and Tables

**Figure 1 antioxidants-13-00149-f001:**
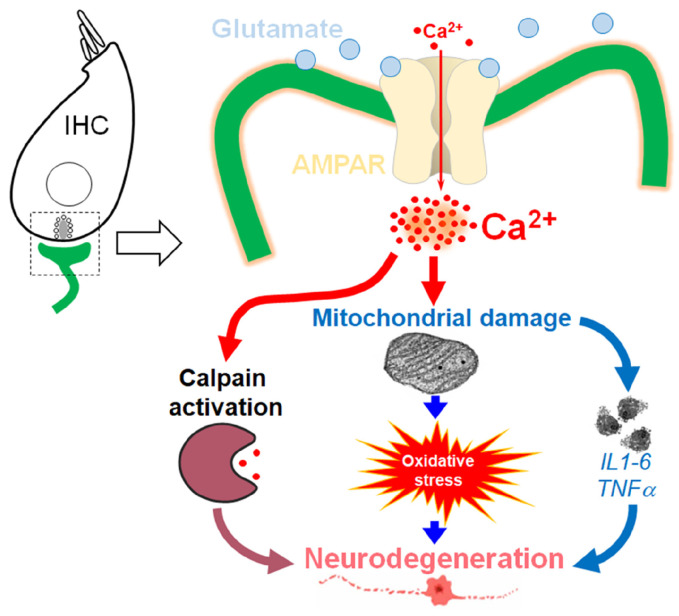
Work hypothesis. Schematic representation of our proposed hypothesis of glutamate excitotoxicity induced by excessive AMPA receptor activation via redundant glutamate release from overstimulated IHCs.

**Figure 2 antioxidants-13-00149-f002:**
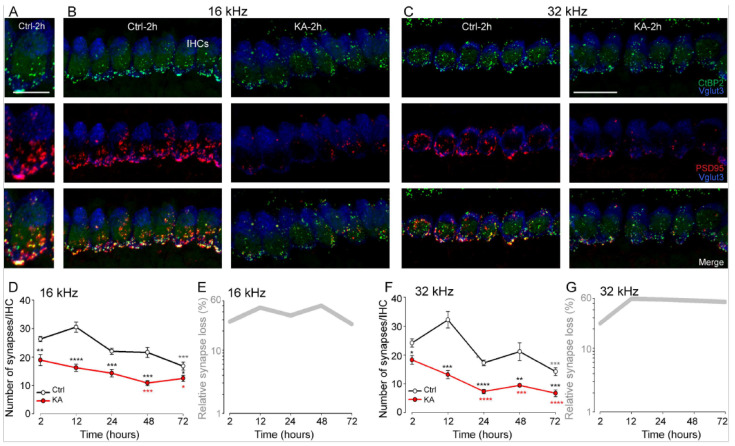
Kainate-induced loss of IHC synapses. (**A**–**C**): Maximum intensity projections of confocal z-stacks of IHCs in the cochlear region coding 16 (**A**,**B**) and 32 (**C**) KHz from the explants treated with either medium alone (Ctrl) or medium containing 0.5 mM kainate for 2 h (KA). The cochleae were labeled with anti-CtBP2 (green), anti-PSD95 (red), and anti-VGLUT3 (blue). Scale bars = 15 µm. IHCs: inner hair cells. (**D**,**F**): KA- and time-dependent loss of the IHC synapses (paired CtBP2-PSD95) in cochlear regions coding 16 KHz (**D**) and 32 KHz (**F**). Control condition (black), KA condition (red) expressed in number of synapses/IHC. (**E**,**G**): Percentage of synapse loss induced by KA compared to control condition calculated over time in cochlear regions coding 16 KHz (**E**) and 32 KHz (**G**). All data are expressed as mean ± SEM (n = nine cochleae per condition). One-way ANOVA test was followed by Dunn’s test: * *p* ≤ 0.05, ** *p* ≤ 0.01, *** *p* ≤ 0.001, **** *p* ≤ 0.0001. Black asterisks: KA vs. Ctrl of the same time point, red asterisks: KA vs. KA 2 h, grey asterisks: Ctrl vs. Ctrl 2 h.

**Figure 3 antioxidants-13-00149-f003:**
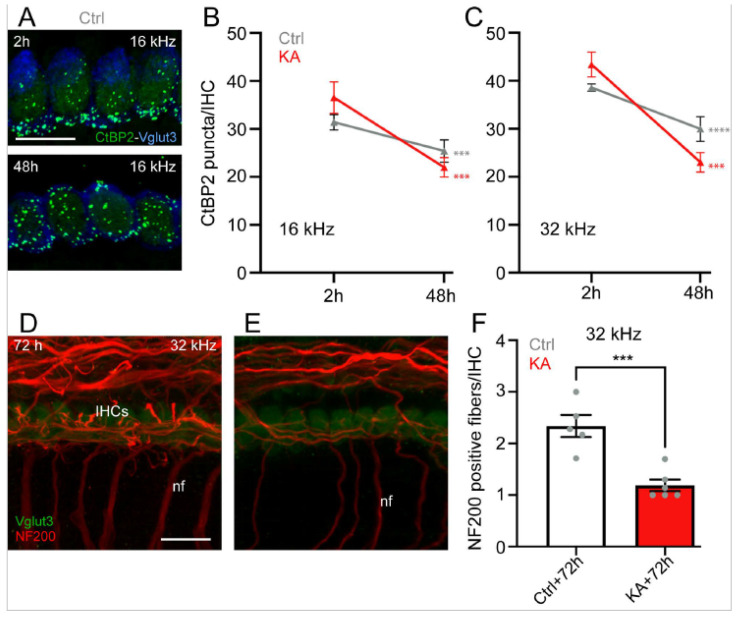
Effects of KA exposure on the number of ribbons and the nerve fiber terminals of the type I spiral ganglion neuron. (**A**): Maximum intensity projections of confocal z-stacks of IHCs in the cochlear region coding 16 KHz from the control explants after 2 h and 48 h in culture. The cochleae were labeled with anti-CtBP2 (green) and anti-VGLUT3 (blue). Scale bars = 15 µm. (**B**,**C**): Quantifications of ribbons (CtBP2 puncta) per IHC in the cochlear region coding 16 and 32 kHz from the control (grey) and KA (red) explants at 2 h and 48 h in culture. Data are expressed as mean ± SEM (n = nine cochleae per condition). *** *p* ≤ 0.001, **** *p* ≤ 0.0001. Red asterisks: KA 48 h vs. KA 2 h, grey asterisks: Ctrl 48 h vs. Ctrl 2 h. (**D**,**E**): Confocal images of the (**D**) control and (**E**) KA-treated cochlear explants maintained in culture for 72 h. The explants were immunolabeled for NF200 (red) and Vglut3 (green), allowing identification of the peripheral processes of the type I spiral ganglion neurons (SGN) and IHCs, respectively. Scale bars = 15 µm. nf: nerve fiber. (**F**): Histogram representing the average number of NF200 positive fibers per IHCs in the regions coding 32 kHz from the KA untreated (Ctrl) and treated (KA) explants 72 h after treatment. Data are expressed as mean ± SEM (n = six cochleae per condition). *** *p* ≤ 0.001. KA vs. Ctrl.

**Figure 4 antioxidants-13-00149-f004:**
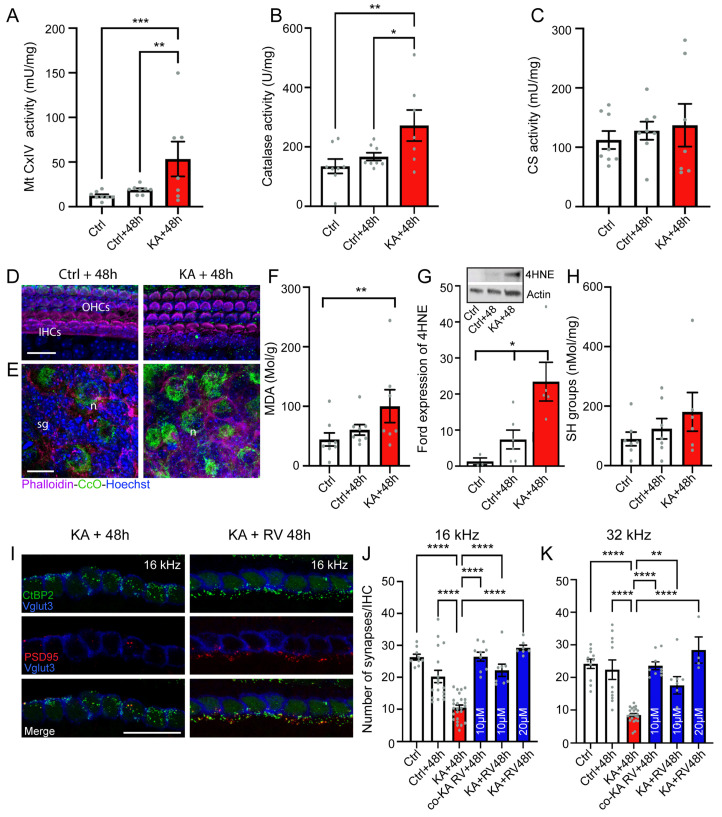
Oxidative stress and antioxidant treatment. (**A**–**C**): Complex IV (CxIV, cytochrome-c oxidase) (**A**), catalase (**B**), and citrate synthase (**C**) activities in whole cochlear extracts from explants incubated with medium alone for 2 h (Crtl), medium alone for 2 h + 48 h (ctrl + 48 h), medium containing kainate for 2 h, and then medium alone for 48 h (KA + 48 h). (**D**,**E**): Confocal images of the organ of Corti (**D**) and SGNs (**E**) from control and KA-treated explants after 48 h in culture. The explants were immunolabeled for cytochrome c oxidase (CoC, green) and counterstained with phalloidin (red) and Hoechst to label actin and nuclei (blue), respectively. OHCs: outer hair cells, IHCs: inner hair cells, sg: spiral ganglion, n: neuron. Scale bars = 15 µm. (**F**,**H**): malondialdehyde (MDA) level (**F**) and Thiols (SH) groups (**H**) in sample extracts from whole cochleae incubated with medium alone for 2 h (Crtl), medium alone for 2 h + 48 h (ctrl + 48 h), medium containing kainate for 2 h, and then medium alone for 48 h (KA + 48 h). Inset in (**G**) and (**G**): Representative Western blot (Inset) and histogram representing the levels of 4HNE in whole cochlear extracts from explants incubated with medium alone for 2 h (Crtl), medium alone for 2 h + 48 h (ctrl + 48 h), medium containing kainate for 2 h, and then medium alone for 48 h (KA + 48 h). β-actin served as a loading control. Data are expressed as mean ± SEM. Each experiment consisted of a pool of 10 cochleae per sample and was performed in biological triplicate and technical replicate. Data are expressed as mean ± SEM for CxIV, catalase, CS, MDA, and SH group analysis. Each experiment consisted of a pool of six cochleae per sample and was performed in eight biological replicates. (**I**): Maximum intensity projections of confocal z-stacks of IHCs in the cochlear region coding 16 KHz from explants treated with either medium containing 0.5 mM KA for 2 h and then medium alone for 48 h (KA + 48 h) or kainate for 2 h and then medium containing resveratrol (RV) for 48 h (KA + RV 48 h). The cochleae were labeled with anti-CtBP2 (green), anti-PSD95 (red), and anti-VGLUT3 (blue). Scale bar = 20 µm. (**J**,**K**): Quantifications of synapses per IHC in cochlear regions coding 16 KHz (**J**) and 32 KHz (**K**) from the explants exposed to medium alone for 2 h (Ctrl), 2 h + 48 h (Ctrl + 48 h); medium containing kainate for 2 h, then medium alone for 48 h (KA + 48 h); medium containing kainate in combination with 10 µM resveratrol for 2 h, then medium alone for 48 h (co-KA RV + 48 h); medium containing kainate for 2 h, then medium containing resveratrol at 10 and 20 µM for 48 h (KA + RV 48 h). All data are expressed as mean ± SEM (n = six to nine cochleae per condition). One-way ANOVA test was followed by Dunn’s test: * *p* ≤ 0.05, ** *p* ≤ 0.01, *** *p* ≤ 0.001, **** *p* ≤ 0.0001.

**Figure 5 antioxidants-13-00149-f005:**
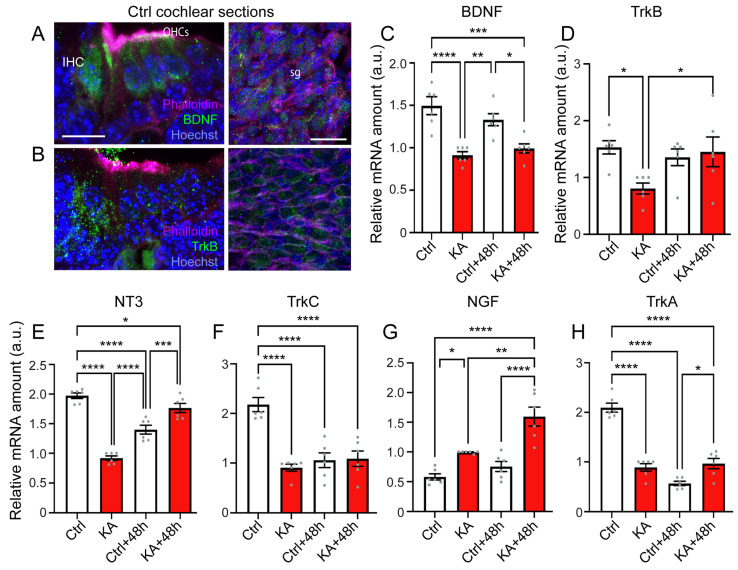
KA induces changes in expression of mRNA-encoding neurotrophins and their receptors. (**A**,**B**): Confocal images of transverse cryostat sections of organ of Corti (**left**) and SGN (**right**) from untreated p3 mouse cochleae. The sections were labeled with anti-BDNF ((**A**), green) and anti-TrkB ((**B**), green), and counterstained with Phalloidin (magenta) and Hoechst (blue) to label actin and nuclei, respectively. OHC: outer hair cell, IHC: inner hair cell, sg: spiral ganglion. Scale bars = 15 µm. (**C**–**H**): Quantitative PCR for *BDNF* (**C**), *TrkB* (**D**), *NT3* (**E**), *TrkC* (**F**), *NGF* (**G**), *TrkA* (**H**), transcripts relative to *Polr2j* and *Ddx48* in whole cochlear extracts from explants incubated with medium alone for 2 h (Ctrl), medium containing kainate for 2 h (KA), medium alone for 2 h + 48 h (Ctrl + 48 h), and medium containing kainate for 2 h and then medium alone for 48 h (KA + 48 h). All data are expressed as mean ± SEM (n = 10 cochleae per sample). All experiments were performed in technical triplicate. One-way ANOVA test was followed by Dunn’s test: * *p* ≤ 0.05, ** *p* ≤ 0.01, *** *p* ≤ 0.001, **** *p* ≤ 0.0001.

**Figure 6 antioxidants-13-00149-f006:**
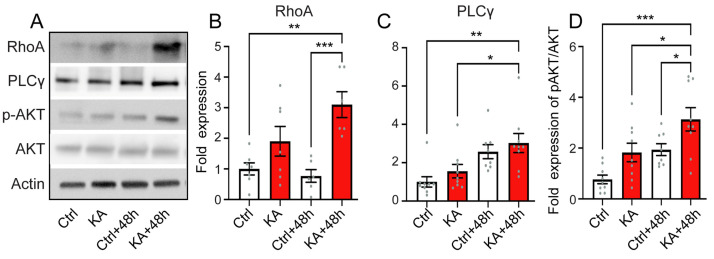
KA induces activation of the downstream pathway of Trk. (A–D): Representative Western blot analysis for RhoA, PLC-γ, p-AKT, and AKT (**A**) and quantification of RhoA (**B**), PLC-γ (**C**), and ratio of p-AKT/AKT (**D**) in sample extracts from whole cochleae incubated with medium alone for 2 h (Ctrl), medium containing kainate for 2 h (KA), medium alone for 2 h + 48 h (Ctrl + 48 h), and medium containing kainate for 2 h and then medium alone for 48 h (KA + 48 h). All data are expressed as mean ± SEM (n = 10 cochleae per sample). All experiments were performed in biological triplicate and two to three technical replicates. β-actin is a leading control. One-way ANOVA test was followed by Dunn’s test: * *p* ≤ 0.05, ** *p* ≤ 0.01, *** *p* ≤ 0.001.

**Figure 7 antioxidants-13-00149-f007:**
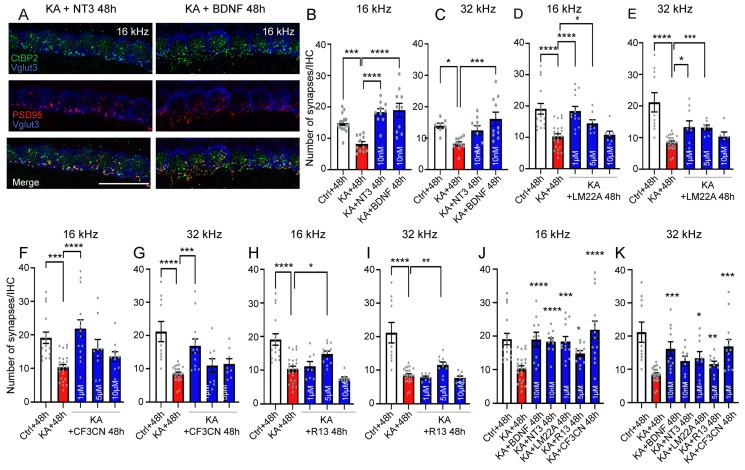
BDNF, NT3, and BDNF mimetics boost synapse regeneration or repair. (**A**–**C**): Maximum intensity projections of confocal z-stacks of IHCs of the cochlear explants labeled with anti-CtBP2 (green), anti-PSD95 (red), and anti-VGLUT3 (blue) (**A**) and quantifications of synapses (paired CtBP2-PSD95) per IHC in cochlear regions coding 16 KHz (**B**) and 32 KHz (**C**). The cochlear samples were imaged in the regions coding 16 KHz and 32 kHz from explants treated with either medium alone for 2 h + 48 h (Ctrl + 48 h), medium containing kainate for 2 h and then medium alone for 48 h (KA + 48 h), medium containing 0.5 mM kainate for 2 h + NT3 for 48 h (KA + NT3 48 h), or kainate for 2 h+ BDNF for 48 h (KA + BDNF 48 h). Scale bar: (**A**) = 15 µm. (**D**–**I**): Quantification of synapses (paired CtBP2-PSD95) per IHC in cochlear regions coding 16 KHz (**D**,**F**,**H**,**J**) and 32 KHz (**E**,**G**,**I**,**K**). In (**D**,**E**), the samples were collected from explants exposed to medium alone for 2 h + 48 h (Ctrl + 48 h); medium containing kainate for 2 h, then medium alone for 48 h (KA + 48 h); medium containing kainate for 2 h, then medium containing LM22A-4 at 1, 5, and 10 µM for 48 h (KA + LM22A-4 48 h). In (**F**,**G**), the samples were collected from explants exposed to medium alone for 2 h + 48 h (Ctrl + 48 h); medium containing kainate for 2 h, then medium alone for 48 h (KA + 48 h); medium containing kainate for 2 h, then medium containing CF3CN at 1, 5, and 10 µM for 48 h (KA + CF3CN 48 h). In (**H**,**I**), the samples were collected from explants exposed to medium alone for 2 h + 48 h (Ctrl + 48 h); medium containing kainate for 2 h, then medium alone for 48 h (KA + 48 h); medium containing kainate for 2 h, then medium containing R13 at 1, 5, and 10 µM for 48 h (KA + R13 48 h). (**J**,**K**): Comparison of the regenerative efficacy of BDNF, NT3, and different BDNF mimetics at their effective concentration in the 16 and 32 kHz coding regions. All data are expressed as mean ± SEM (n = 6–10 cochleae per condition). One-way ANOVA test was followed by Dunn’s test: * *p* ≤ 0.05, ** *p* ≤ 0.01, *** *p* ≤ 0.001, **** *p* ≤ 0.0001.

**Figure 8 antioxidants-13-00149-f008:**
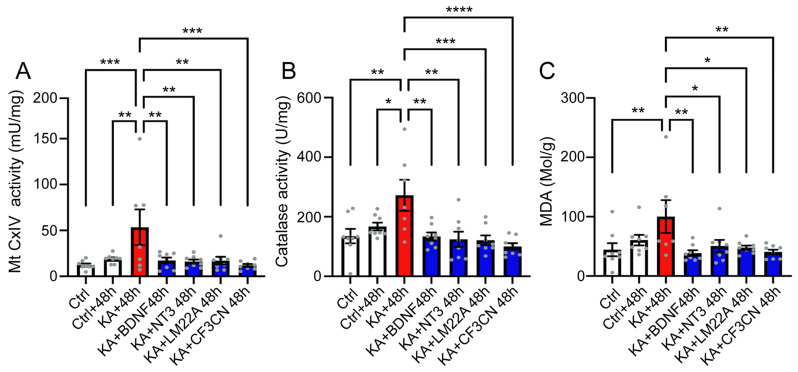
Trk B agonist treatment reduces oxidative stress. (**A**–**C**): CxIV (**A**) and catalase (**B**) activities and MDA levels (**C**) in sample extracts from whole cochleae incubated with medium alone for 2 h (Ctrl); medium alone for 2 h + 48 h (Ctrl + 48 h); medium containing kainate for 2 h, then medium alone for 48 h (KA + 48 h); medium containing kainate for 2 h, then medium containing BDNF (10 nM) or NT3 (10 nM) or LM22A-4 (1 µM) or CF3CN (1 µM) for 48 h. All data are expressed as mean ± SEM (each experiment consisted of a pool of six cochleae per sample and was performed in eight biological replicates). One-way ANOVA test was followed by Dunn’s test: * *p* ≤ 0.05, ** *p* ≤ 0.01, *** *p* ≤ 0.001, **** *p* ≤ 0.0001.

**Table 1 antioxidants-13-00149-t001:** Sequences of the primer pairs used in this study. F: forward primer, R: reverse primer.

Gene	5′ to 3′ Primer	Nucleotide Position	Product Size (bp)	GenBank Accession Number
Bdnf	F-AAAGTCCCGGTATCCAAAGG R-ATCGCCAGCCAATTCTCTTT	879–1062	184	NM_001048142
Ngf	F-GCAGTGAGGTGCATAGCGTA R-CTGTGTCAAGGGAATGCTGA	286–444	159	NM_013609
Nt3	F-TCTGCCACGATCTTACAGGTG R-AGGGTGCTCTGGTAATTTTCCT	304–521	218	NM_001164034
TrkA	F-GAACCCACTGCATTGTTCCT R-GCACTGCAGAAACACGTCAT	459–669	211	NM_001033124
TrkB	F-ACTGTCCTGCTACCGCAGTT R-GTTCACAGTGGCTGGGACAT	289–474	186	NM_001025074
TrkC	F-CCTGACACAGTGGTCATTGG R-CTTGTCTTTGGTGGGGCTTA	1547–1753	207	NM_008746
Ddx48	F-GGAGTTAGCGGTGCAGATTC R-AGCATCTTGATAGCCCGTGT	395–598	204	NM_138669
Polr2j	F-ACCACACTCTGGGGAACATC R-CTCGCTGATGAGGTCTGTGA	176–353	178	NM_011293

## Data Availability

The data are contained within the article.

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
