# Peer review of "Oxidative Stress Plays an Important Role in Glutamatergic Excitotoxicity-Induced Cochlear Synaptopathy: Implication for Therapeutic Molecules Screening"

_antioxidants, 2024, doi:10.3390/antiox13020149_

Round 1
Reviewer 1 Report
Comments and Suggestions for Authors
P3 mouse cochlear explants were exposed to KA for 2 h to cause excitotoxic damage, then maintained in medium for 0, 12, 24, 48 and 72 h post KA-exposure. Measurements of synapses per IHC were made from KA-exposed explants and from untreated control explants at each time point. Progressive reduction of synapses was observed in both KA-treated and control explants, with significantly greater loss in KA-exposed explants in 16- and 32 kHz regions of the cochlea. Increased levels of oxidative stress and altered neurotrophin signaling pathways and mitochondrial function were observed in KA-exposed samples. Reserveratrol and neurotrophins had beneficial effects.
This is a timely, comprehensive, and well-executed set of studies with potential clinical implications for future treatment of cochlear synaptopathies associated with ototoxicity, noise exposure, and aging. The methods are appropriate and thorough, and the results are clear and convincing. I have no major criticisms or concerns, only a very minor suggestion regarding statistical presentation. Throughout the results section, p values are presented as if they represent effect sizes, which they do not. Please consider providing effect size measures along with p values—this would show readers how strong/big the effects are in addition to whether or not they were significant (i.e., unlikely to happen if the null hypothesis is true and there is no effect of treatment).
The suggestions below pertain to improving grammar and clarity of presentation:
L34, dendritic processes extend toward hair cells, not just IHCs; L35 delete “central” (both occurrences) as the axonic process extends centrally but is itself a peripheral process. So “a dendritic process extending toward the hair cells of the organ of Corti and an axonic process that extends centrally…”
--But I think this and the abstract are the only places in the manuscript that you refer to a “dendritic process”; you might want to use terminology consistently throughout the paper
-L36, change to 90-95% to be consistent with L40, 5-10% of SGNs being Type II
-L65, noise-induced
-L73 involved in
-L79, I think you mean efficacy, not efficiency?
-L193, per sample
-L231, fix: “distal peripheral axons of type I SGN peripheral axons” (and note, is this what you called dendrites in the Introduction?”)
-L249; L316, L424, L513: Four incomplete sentences that start with, “Whereas…”
-L361, were significantly increased 48 h
-L439, may be involved
-L470, run-on sentence
-L458, “Since…” incomplete sentence; also, “promote…against” is not grammatically correct.
-L 486, L489, L492 “samples were connected from explants”—Change all instances of “connected” to “collected”
-L505, cochleae per sample
-L539, If this is the case
-L581, already shown
Comments on the Quality of English LanguageThe suggestions below pertain to improving grammar and clarity of presentation:
L34, dendritic processes extend toward hair cells, not just IHCs; L35 delete “central” (both occurrences) as the axonic process extends centrally but is itself a peripheral process. So “a dendritic process extending toward the hair cells of the organ of Corti and an axonic process that extends centrally…”
--But I think this and the abstract are the only places in the manuscript that you refer to a “dendritic process”; you might want to use terminology consistently throughout the paper
-L36, change to 90-95% to be consistent with L40, 5-10% of SGNs being Type II
-L65, noise-induced
-L73 involved in
-L79, I think you mean efficacy, not efficiency?
-L193, per sample
-L231, fix: “distal peripheral axons of type I SGN peripheral axons” (and note, is this what you called dendrites in the Introduction?”)
-L249; L316, L424, L513: Four incomplete sentences that start with, “Whereas…”
-L361, were significantly increased 48 h
-L439, may be involved
-L470, run-on sentence
-L458, “Since…” incomplete sentence; also, “promote…against” is not grammatically correct.
-L 486, L489, L492 “samples were connected from explants”—Change all instances of “connected” to “collected”
-L505, cochleae per sample
-L539, If this is the case
-L581, already shown
Author Response
We would like to thank the editor and reviewers for their thoughtful comments and efforts towards improving our manuscript. Our revisions reflect all reviewers’ suggestions and comments. Detailed responses to reviewers are given below.
Reviewer 1
- Reviewer comments
P3 mouse cochlear explants were exposed to KA for 2 h to cause excitotoxic damage, then maintained in medium for 0, 12, 24, 48 and 72 h post KA-exposure. Measurements of synapses per IHC were made from KA-exposed explants and from untreated control explants at each time point. Progressive reduction of synapses was observed in both KA-treated and control explants, with significantly greater loss in KA-exposed explants in 16- and 32 kHz regions of the cochlea. Increased levels of oxidative stress and altered neurotrophin signaling pathways and mitochondrial function were observed in KA-exposed samples. Reserveratrol and neurotrophins had beneficial effects.
This is a timely, comprehensive, and well-executed set of studies with potential clinical implications for future treatment of cochlear synaptopathies associated with ototoxicity, noise exposure, and aging. The methods are appropriate and thorough, and the results are clear and convincing. I have no major criticisms or concerns, only a very minor suggestion regarding statistical presentation.
Authors' answer
We are grateful to the reviewer for these positive comments, careful reading of our manuscript, and valuable and constructive suggestions. These insights helped us to improve the quality of the manuscript.
- Reviewer comments
Throughout the results section, p values are presented as if they represent effect sizes, which they do not. Please consider providing effect size measures along with p values—this would show readers how strong/big the effects are in addition to whether or not they were significant (i.e., unlikely to happen if the null hypothesis is true and there is no effect of treatment).
Authors' answer
We have taken the reviewer’s comments into consideration and have added size measures along with p values.
See the results section.
- Reviewer comments
The suggestions below pertain to improving grammar and clarity of presentation:
L34, dendritic processes extend toward hair cells, not just IHCs; L35 delete “central” (both occurrences) as the axonic process extends centrally but is itself a peripheral process. So “a dendritic process extending toward the hair cells of the organ of Corti and an axonic process that extends centrally…”
--But I think this and the abstract are the only places in the manuscript that you refer to a “dendritic process”; you might want to use terminology consistently throughout the paper.
Authors' answer
We want to thank the Reviewer for his constructive remarks and comments. All these issues have been addressed in this revised version of the manuscript.
- Reviewer comments
L36, change to 90-95% to be consistent with L40, 5-10% of SGNs being Type II
Authors' answer
Done, thanks!
- Reviewer comments
-L65, noise-induced
-L73 involved in
-L79, I think you mean efficacy, not efficiency?
-L193, per sample
Authors' answer
All these issues have been addressed in the revised version of the manuscript, thanks!
- Reviewer comments
L231, fix: “distal peripheral axons of type I SGN peripheral axons” (and note, is this what you called dendrites in the Introduction?”)
Authors' answer
As recommended by the reviewer, we have standardized our language by replacing “dendritic process” with “peripheral process” throughout the paper.
- Reviewer comments
-L249; L316, L424, L513: Four incomplete sentences that start with, “Whereas…”
Authors' answer
These issues have been addressed, we apology for any errors in using inappropriate words to introduce opposing notions.
See line 293-296.
“In contrast, the similar number of ribbons/IHC was observed in both control and KA exposed cochlear explant after two hours of treatment (Figure 3A-C). Furthermore, IHCs, OHCs, and SGNs seem intact up to 48 h in culture (data not shown).”
L363-364.
“On the other hand, no significant difference in the level of CS activity was found between groups (Figure 4C).”
L488-492
“Although, significant recovery of NT3 expression was seen 48 h after KA exposure compared with control cochleae 48 h in culture (p⩽0.001, n=6 qPCR from 2 RT reaction), at this time point, NT3 level was still lower than control at 2 h in culture (p⩽0.05, n=6 qPCR from 2 RT reaction, Figure 5E).”
L609.
“In contrast, the death of the cell bodies of the SGNs occurs over years in mice”.
- Reviewer comments
L361, were significantly increased 48 h
-L439, may be involved
-L470, run-on sentence
-L458, “Since…” incomplete sentence; also, “promote…against” is not grammatically correct.
-L 486, L489, L492 “samples were connected from explants”—Change all instances of “connected” to “collected”
-L505, cochleae per sample
-L539, If this is the case
-L581, already shown
Authors' answer
These issues have been addressed, thank you very much.
- Reviewer comments
The suggestions below pertain to improving grammar and clarity of presentation:
L34, dendritic processes extend toward hair cells, not just IHCs; L35 delete “central” (both occurrences) as the axonic process extends centrally but is itself a peripheral process. So “a dendritic process extending toward the hair cells of the organ of Corti and an axonic process that extends centrally…”
--But I think this and the abstract are the only places in the manuscript that you refer to a “dendritic process”; you might want to use terminology consistently throughout the paper
-L36, change to 90-95% to be consistent with L40, 5-10% of SGNs being Type II
-L65, noise-induced
-L73 involved in
-L79, I think you mean efficacy, not efficiency?
-L193, per sample
-L231, fix: “distal peripheral axons of type I SGN peripheral axons” (and note, is this what you called dendrites in the Introduction?”)
-L249; L316, L424, L513: Four incomplete sentences that start with, “Whereas…”
-L361, were significantly increased 48 h
-L439, may be involved
-L470, run-on sentence
-L458, “Since…” incomplete sentence; also, “promote…against” is not grammatically correct.
-L 486, L489, L492 “samples were connected from explants”—Change all instances of “connected” to “collected”
-L505, cochleae per sample
-L539, If this is the case
-L581, already shown
Authors' answer
We would like to thank the reviewer for her/his thoughtful comments and efforts towards improving our manuscript.
All these issues have been addressed in this revised version of the manuscript, thanks!

Reviewer 2 Report
Comments and Suggestions for Authors
I appreciate the opportunity to review this manuscript. Using murine cochlear explants, the authors focused on the role of kainate receptors in mediating cochlear synaptopathy. They demonstrated that the levels of neurotrophins and their receptors are altered by kainate at the transcript level. They also provided evidence for the induction of oxidative stress by kainate. Finally, they showed that exogenous neurotrophins counteract kainite-induced synaptopathy and reduce oxidative stress.
In the Introduction, the authors mention that the spiral ganglion neurons are bipolar. Only type I SGN (innervating IHCs) are bipolar, whereas type II (innervating OHCs) are pseudounipolar. Please correct this information (see Carricondo, F. and Romero-Gómez, B. (2019), The Cochlear Spiral Ganglion Neurons: The Auditory Portion of the VIII Nerve. Anat. Rec., 302: 463-471. https://doi.org/10.1002/ar.23815).
Lines 56-57 – “glutamate/aspartate transporter protein (GLAST), responsible for the uptake of glutamate GLAST” – please delete the second “GLAST” as it does not make sense.
Lines 68 – 69 “While it has been demonstrated that AMPA receptor over activation is sufficient to cause synaptopathy, the excitotoxic mechanisms of glutamate in the cochlea are unknown”. This is a strong statement that does not reflect the truth. Not everything is known about the mechanism of excitotoxicity in the cochlea, but a lot of information has been published, e.g.,:
Negandhi J, Harrison AL, Allemang C, Harrison RV. Time course of cochlear injury discharge (excitotoxicity) determined by ABR monitoring of contralateral cochlear events. Hear Res. 2014 Sep;315:34-9. doi: 10.1016/j.heares.2014.06.002. Epub 2014 Jun 26. PMID: 24973579.
Moser T, Starr A. Auditory neuropathy--neural and synaptic mechanisms. Nat Rev Neurol. 2016 Mar;12(3):135-49. doi: 10.1038/nrneurol.2016.10. Epub 2016 Feb 19. PMID: 26891769.
Moser T, Predoehl F, Starr A. Review of hair cell synapse defects in sensorineural hearing impairment. Otol Neurotol. 2013 Aug;34(6):995-1004. doi: 10.1097/MAO.0b013e3182814d4a. PMID: 23628789.
Line 76 “to inform the work to develop effective therapies” – please revise
Line 84: “Neonate Swiss mice were purchased from Janvier Laboratories (Le Genest-Saint-Isle) and were housed in pathogen-free animal care facilities.” – this is a confusing sentence. Were the pups purchased and transported to the local animal facility where they were housed without the mother? How long were they housed there? How many mice in total were used? What was the sex of the mice used?
Please describe how the physical location in the cochlea was determined for each tested frequency.
Line 139: “In control specimens without primary antibodies, neither Alexa 488, 568 nor 647 fluorescent tags were observed.” This sentence is part of the results. Please state here (in MM) how the control samples were treated/stained without stating the results.
General note for MM: Please provide a table with information about the antibodies/fluorescent dyes used in the experiments. Such a table should include the target's name, the antibody used, the species from which the antibody was derived, the isotype, the catalog number, the company, and the dilution factor.
Please precisely specify the parameters used during confocal imaging.
Lines 166-167: “Each experiment consisted of a pool of 6 cochleae per samples and were performed in 8 biological replicates”. Please specify if the pools were prepared from 6 cochleae from 3 or 6 animals.
Lines 174-175: Please indicate that phalloidin was used to visualize filamentous actin.
Paragraph 2.9 (Immunoblotting) – Please describe the conditions of PAGE (which buffers were used, which gels and of what percentage, gradient or not, time of the run, voltage, device, manufacturer, etc.). Same for the blotting procedure. What membrane was used? How was the signal from WB acquired? – please provide the device's name, type, and manufacturer. Moreover, please specify if the pools were prepared from 10 cochleae from 5 or 10 animals.
Paragraph 2.10 TRIZOL (or TRI-reagent) is a reagent, not a method. Please describe the methods used in this work or at least provide a reference. Please describe how the RNA was quantified and what was used to control the quality of RNA.
Please provide the base number (on a target cDNA) and the amplicon size when listing the primers. Also, here, I would advise using a tabular format.
“All experiments were performed in technical triplicate.” – how many cochleae were analyzed individually? Or was the RNA/cDNA pooled?
Figure 1 The nuclei and the VGLUT3 were visualized in the blue channel. Could the authors use a false color for VGLUT3? This extends to all images where two blue dyes were used.
Figure 3A This figure deserves a separate presentation and description, as it presents the authors' working hypothesis. I suggest placing it in the Introduction or the Discussion.
Lines 319-320: “…showing stronger cytochrome c oxidase expression in the cytoplasm of the SGNs 48h after KA exposure when compared with control unexposed cochlear explants” – how was the “stronger expression” quantified?
“We observed that MDA levels were significantly increase in 48 h after KA exposure (p⩽0.01, Figure 3G), while, SH levels were only slightly, but not significantly, increased 48 h after KA exposure compared with control unexposed cochleae (Figure 3I).” What do these result suggest?
Figure 4: The title “Neurotrophic signaling pathway under excitotoxicity” is misleading, as no signaling was studied here (apart from AKT phosphorylation, which is 8% of the figure). Please consider revising it into a more descriptive title (e.g., “KA induces changes in expression of mRNA encoding neurotrophins and their receptors”). Moreover, split this busy figure into a part belonging to paragraph 4.5 and the other – 4I, 4J, 4K, and 4L) belonging to paragraph 4.6. The axis in figures 4 C, D, E, F, and H are labeled with “relative mRNA amount (a.u.)”. I take it that “a.u.” stands for “arbitrary units”. How were these units derived/calculated? This should be described in paragraph 2.10.
Paragraph 4.6 – The authors talk here about “protein expression”; however, using the WB, not the expression but the levels of proteins were studied. Please revise.
Paragraph 4.7 – “We therefore investigated the efficiency of exogenous BDNF and NT3 to boost synapse regeneration.” Using the term “synapse regeneration” requires additional experiments, which were not performed. The authors studied changes in the number of synapses, which could result from synaptic regeneration, de novo synaptogenesis, inhibition of synaptopathy, or all of the above.
Figure 5A – the images are illegible. Please consider using larger micrographs.
Lines 560-561 – “Together, these results strongly support our hypothesis that glutamatergic excitotoxicity-induced loss of synapses is mediated by increased production of ROS and oxidative stress” – I suggest communicating this more cautiously, e.g., “(…) is mediated at least in part by increased production….”.
Comments on the Quality of English LanguageThe syntax and grammar need polishing.
Author Response
- Reviewer comments
I appreciate the opportunity to review this manuscript. Using murine cochlear explants, the authors focused on the role of kainate receptors in mediating cochlear synaptopathy. They demonstrated that the levels of neurotrophins and their receptors are altered by kainate at the transcript level. They also provided evidence for the induction of oxidative stress by kainate. Finally, they showed that exogenous neurotrophins counteract kainite-induced synaptopathy and reduce oxidative stress.
Authors' answer
We are grateful to the reviewer for these positive comments and constructive suggestions for improving our work.
- Reviewer comments
In the Introduction, the authors mention that the spiral ganglion neurons are bipolar. Only type I SGN (innervating IHCs) are bipolar, whereas type II (innervating OHCs) are pseudounipolar. Please correct this information (see Carricondo, F. and Romero-Gómez, B. (2019), The Cochlear Spiral Ganglion Neurons: The Auditory Portion of the VIII Nerve. Anat. Rec., 302: 463-471. https://doi.org/10.1002/ar.23815).
Authors' answer
This information has been corrected, we have also cited the reference above. Thank you very much for your helpful review!
See L35-40
“Spiral ganglion neurons (SGNs) of the cochlea are composed of two types of the sensory neurons. 90-95% of the SGNs are type I afferent neurons, which are myelinated and synapse on inner hair cells (IHCs). Each terminal of type I SGNs innervates only one IHC, while each IHC receives contacts from 10 to 20 terminals of type I SGNs [1]. Type II SGNs represent only 5 to 10% of the afferent neurons [2,3] and are non-myelinated and pseudounipolar [4–6].”
- Reviewer comments
Lines 56-57 – “glutamate/aspartate transporter protein (GLAST), responsible for the uptake of glutamate GLAST” – please delete the second “GLAST” as it does not make sense.
Authors' answer
Done, thanks.
- Reviewer comments
Lines 68 – 69 “While it has been demonstrated that AMPA receptor over activation is sufficient to cause synaptopathy, the excitotoxic mechanisms of glutamate in the cochlea are unknown”. This is a strong statement that does not reflect the truth. Not everything is known about the mechanism of excitotoxicity in the cochlea, but a lot of information has been published, e.g.,:
Authors' answer
We fully agree with the reviewer that this statement was too strong. We have replaced “unknown” by “not completely understood”.
See page L68-70
“While it has been demonstrated that AMPA receptor over activation is sufficient to cause synaptopathy, the excitotoxic mechanisms of glutamate in the cochlea are not completely understood.”
- Reviewer comments
Line 76 “to inform the work to develop effective therapies” – please revise
Done, thanks
- Reviewer comments
Line 84: “Neonate Swiss mice were purchased from Janvier Laboratories (Le Genest-Saint-Isle) and were housed in pathogen-free animal care facilities.” – this is a confusing sentence. Were the pups purchased and transported to the local animal facility where they were housed without the mother? How long were they housed there? How many mice in total were used? What was the sex of the mice used?
Authors' answer
Done, thanks.
See L102-L109.
“100 pregnant female Swiss mice were purchased from Janvier Laboratories (Le Genest-Saint-Isle) and were housed in pathogen-free animal care facilities accredited by the French Ministry of Agriculture and Food (D-34-172-36, May 20, 2021). Each pregnant mouse gives birth to litters of around ten babies on average. Experiments were carried out on third-day neonatal mouse pups (P3) without distinguishing their sex. All protocols comply with French Ethical Committee stipulations regarding the care and use of animals for experimental procedures (agreements C75-05-18 and 01476.02, license #6711). All efforts were made to minimize the number of animals used.”
- Reviewer comments
Please describe how the physical location in the cochlea was determined for each tested frequency.
Authors' answer
Done, thanks.
See L165-169
“The quantitative analyses (6 to 10 cochleae per condition and per time point) were performed in over 10 successive IHCs in the cochlear regions centered at 1.1, 2.6 or 4.1 mm from the cochlear apical end and corresponding to the frequencies of 8, 16 and 32 kHz, respectively [47].”
- Reviewer comments
Line 139: “In control specimens without primary antibodies, neither Alexa 488, 568 nor 647 fluorescent tags were observed.” This sentence is part of the results. Please state here (in MM) how the control samples were treated/stained without stating the results.
Authors' answer
As recommended by the reviewer, we have added more detailed information on treatments control samples and negative controls.
Please see L146-L147.
“All control samples were maintained in culture medium alone and were run concurrently alongside the experimental cultures.”
L159-L161
“To exclude nonspecific binding of the secondary antibodies, negative controls were performed without primary antibodies.”
- Reviewer comments
General note for MM: Please provide a table with information about the antibodies/fluorescent dyes used in the experiments. Such a table should include the target's name, the antibody used, the species from which the antibody was derived, the isotype, the catalog number, the company, and the dilution factor.
Please precisely specify the parameters used during confocal imaging.
Authors' answer
To take into account the reviewer's comments, we have added two new Supplementary Tables including all requested information, thank you for the helpful reviewer comments.
See new Supplementary table 1 and 2
Also see L161-165.
“Fluorescent tags were visualized using confocal microscopy (Zeiss 880 Airyscan) with a Plan-Apochromat 63X/1,4 Oil DIC M27 objective and imaged with a 4-channel z-stack spanning the height of the hair cells to capture all synaptic puncta. The confocal images were acquired with 1024 pixels × 512 pixels in x and y, z- spacing at 0.32 µ m per slice (Table S2).”
- Reviewer comments
Lines 166-167: “Each experiment consisted of a pool of 6 cochleae per samples and were performed in 8 biological replicates”. Please specify if the pools were prepared from 6 cochleae from 3 or 6 animals.
Authors' answer
Done, thanks.
See L193-194
“Each experiment consisted of a pool of 6 cochleae from 3 mice per sample and were performed in 8 biological replicates.”.
- Reviewer comments
Lines 174-175: Please indicate that phalloidin was used to visualize filamentous actin.
Authors' answer
Perfomed, thanks.
Please see L201-L203
“The samples were counterstained with Alexa 647 phalloidin (1:1000, Thermo Fisher Scientific, #A22287) to visualize filamentous actin, and Hoechst 33342 (1:5000, Thermo Fisher Scientific, #62249) to stain the nuclei.”.
- Reviewer comments
Paragraph 2.9 (Immunoblotting) – Please describe the conditions of PAGE (which buffers were used, which gels and of what percentage, gradient or not, time of the run, voltage, device, manufacturer, etc.). Same for the blotting procedure. What membrane was used? How was the signal from WB acquired? – please provide the device's name, type, and manufacturer. Moreover, please specify if the pools were prepared from 10 cochleae from 5 or 10 animals.
Authors' answer
To address the reviewer's comment, we have now provided a more detailed immunoblotting protocol. Thank you for the constructive comments of the reviewer.
See L210-L242.
“Lysate of the cochlear samples was obtained using RIPA buffer (Thermofisher #89900). The protein concentration was evaluated by Pierce™ BCA Protein Assay Kits (Thermofisher #23225). Cochlear homogenates were prepared in 4X Laemmli sample buffer (Biorad #1610747) for a ratio of ¼ protein/Laemmli sample buffer. The protein was then denatured for 5 minutes at 95° C. 15 µg of protein was loaded per lane in a pre-made gel (Biorad, Any kD MP TGX Stain-Free 10W 50 μl pkg 10 #4568124). The electrophoresis apparatus (Biorad) was filled with 1X Tris/glycine/SDS running buffer to separate protein samples by SDS-PAGE (Biorad, #1610772). The gel was run at 90 V for 10 min then 120 V for 50 min. The protein transfer was performed using PVDF membrane (Biorad, Trans-Blot Turbo Midi PVDF Transfer Packs #1704157) in Biorad apparatus (Trans-Blot® Turbo™ Transfer System #1704150) according to the manufacturer’s instructions.
The membranes were rinsed and then incubated in blocking solution (milk 5% in 1X TBST) for 1 hour at room temperature. Next, they were incubated with primary antibodies overnight at 4°C in 1X TBST with 5% milk. The membrane was washed three times with TBST (purchased from Fisher Bio regeant #BP2471-1, TWEEN® 20 from Sigmaldrich #P5927). The primary antibodies used were: antibodies recognizing 4-Hydroxynonenal (4HNE, 1:1000, Bioss Antibodies, #bs-6313R RRID:AB_2827741), Ras homolog gene family member A (RhoA , 1:1000, Cell Signaling, #2117 RRID:AB_10693922), phospholipase C-γ (PLC-γ, 1:1000, Cell Signaling, #2822 RRID: AB_2163702), AKT (1:1000, Cell Signaling, #4691 RRID:AB_915783), pAKT (1:1000, Cell Signaling, #4060 RRID:AB_2315049). β-actin (1:10000, Sigma-Aldrich, #A1978 RRID: AB-476692) served as a loading control.
The membranes were then incubated with the secondary antibodies for 2 hours at room temperature. The secondary antibodies used were horseradish peroxidase-conjugated goat anti-mouse IgG (1:3000, Jackson ImmunoResearch, #115-001-003 RRID: AB-2338443), or goat anti-rabbit IgG (1:3000, Jackson ImmunoResearch, #111-001-003 RRID: AB-2337910). Finally, the membranes were washed three times with TBST. The detection of protein was performed using chemiluminescent reagent (Thermo Fisher scientific, SuperSignal™ West Pico PLUS Chemiluminescent Substrate, #34580) and Vilber fusion Fx device.
Image scans of the Western blots were used for semi-quantitative analysis with Fiji software. Each experiment consisted of a pool of 10 cochleae from 5 mice per sample and was performed in biological triplicate and 2-3 technical replicate. All results were normalized by β-actin expression.”
- Reviewer comments
Paragraph 2.10 TRIZOL (or TRI-reagent) is a reagent, not a method. Please describe the methods used in this work or at least provide a reference. Please describe how the RNA was quantified and what was used to control the quality of RNA.
Please provide the base number (on a target cDNA) and the amplicon size when listing the primers. Also, here, I would advise using a tabular format.
“All experiments were performed in technical triplicate.” – how many cochleae were analyzed individually? Or was the RNA/cDNA pooled?
Authors' answer
All these issues have been addressed according to the reviewer’s comments and suggestions, thanks you very much for your excellent review and editing work.
See L244-L259
“Total RNA was extracted from 10 cochleae per sample using TRI-reagent (Invitrogen, # 9738G) according to the manufacturer's instruction. Concentration and purity were assessed using a NanoDrop spectrophotometer with the ratios A260/A280 and A260/A230. Samples were reverse transcribed using PrimeScrip RT Reagent Kit (Qiagen, #330401). Real-time PCR was carried out using SYBR Green I dye detection on the Light Cycler system (Roche Molecular Biochemicals). PCR reactions were carried out in 96-well plates in a 10 µl volume containing 3 µl of cDNA product (final dilution 1/30), 0.5 µM of forward and reverse primers, and 2 µl of QuantiTect SYBR Green PCR Master Mix (Roche Diagnosis). Sequences of the primer pairs used are listed in Table 1. All experiments were performed in technical triplicate on two independent RT reaction products. The relative amounts of specifically amplified cDNAs calculated on at least three independent experimental replicates using the delta-CT method [55] and were normalized with polymerase (RNA) II polypeptide J, Polr2j and DEAD box polypeptide 48, Ddx48 as stable control genes. In particular, arbitrary units (a.u.) of each gene expression was calculated through normalization with a factor. This normalization factor represents the geometric averaging of the expression of two stable control genes as described by Vandesompele et al [55].”
- Reviewer comments
Figure 1 The nuclei and the VGLUT3 were visualized in the blue channel. Could the authors use a false color for VGLUT3? This extends to all images where two blue dyes were used.
Authors' answer
We are sorry for this confusion, however, in the images shown in Figure 1 we did not show the Hoechst-stained nuclei, since CtBP2 labeled presynaptic ribbons and IHC nuclei at the same time, shown here with false colors in greens.
- Reviewer comments
Figure 3A This figure deserves a separate presentation and description, as it presents the authors' working hypothesis. I suggest placing it in the Introduction or the Discussion.
Authors' answer
Done, thanks.
See L75-L83 and new Figure 1.
“Here, our working hypothesis is that overstimulation of IHCs by noise may lead to massive calcium influx which can induce activation of the calpain pathway and cause mitochondrial damage and subsequent increase in levels of oxidative stress and pro-inflammatory process. Together, this leads to neurodegeneration of the auditory nerve terminals and synaptopathy (Figure 1). To test this hypothesis, we examined the molecular basis responsible for KA-induced loss of IHC synapses and degeneration of the auditory nerve terminals of type I SGNs using a cochlear explant culture from P3 mouse pups. In addition, we assessed the efficacy of antioxidant and neurotrophic factors, such as NT3, BDNF, and some mimetics of BDNF, on cochlear synapse regeneration.”
And also Figure 1 legend.
“Figure 1. Work hypothesis. Schematic representation of our proposed hypothesis of glutamate excitotoxicity induced by excessive AMPA receptor activation via redundant glutamate release from overstimulated IHCs.”
- Reviewer comments
Lines 319-320: “…showing stronger cytochrome c oxidase expression in the cytoplasm of the SGNs 48h after KA exposure when compared with control unexposed cochlear explants” – how was the “stronger expression” quantified?
Authors' answer
Agree, we have accordingly modified this sentence.
See L364-366.
“The increase in the activity of mt CxIV was consistent with confocal microscopy observations showing intense expression of cytochrome c oxidase mainly in the cytoplasm of the SGNs 48 h after KA exposure (Figure 4D-E).”
- Reviewer comments
“We observed that MDA levels were significantly increase in 48 h after KA exposure (p⩽0.01, Figure 3G), while, SH levels were only slightly, but not significantly, increased 48 h after KA exposure compared with control unexposed cochleae (Figure 3I).” What do these result suggest?
Authors' answer
Done, thanks.
See L414-418.
“We observed that MDA levels were significantly increased in KA exposed cochleae 48 h after exposure compared with control unexposed cochleae (p⩽0.01, n= 8 biological replicates, Figure 4F), while, no significant difference in SH levels was seen between KA exposed and unexposed cochleae (Figure 4H). These results suggest that KA induced lipid peroxidation.”
- Reviewer comments
Figure 4: The title “Neurotrophic signaling pathway under excitotoxicity” is misleading, as no signaling was studied here (apart from AKT phosphorylation, which is 8% of the figure). Please consider revising it into a more descriptive title (e.g., “KA induces changes in expression of mRNA encoding neurotrophins and their receptors”). Moreover, split this busy figure into a part belonging to paragraph 4.5 and the other – 4I, 4J, 4K, and 4L) belonging to paragraph 4.6. The axis in figures 4 C, D, E, F, and H are labeled with “relative mRNA amount (a.u.)”. I take it that “a.u.” stands for “arbitrary units”. How were these units derived/calculated? This should be described in paragraph 2.10.
Authors' answer
As recommended by the reviewer, we have divided Figure 4 into new Figures 5 and 6. We have revised and changed the titles of the new Figures 5 and 6. We have also given the description of a.u. in paragraph 2.10.
Thank you for the reviewer’s excellent reviewing and editing work.
Please note that we found an error in the primer sequences listed for NT3 in the previous version of the manuscript. This error has been corrected in Table 1
See new figures 5 and 6 and their titles and new Table 1.
Also see L257-L259.
“In particular, arbitrary units (a.u.) of each gene expression was calculated through normalization with a factor. This normalization factor represents the geometric averaging of the expression of two stable control genes as described by Vandesompele et al [55].”
- Reviewer comments
Paragraph 4.6 – The authors talk here about “protein expression”; however, using the WB, not the expression but the levels of proteins were studied. Please revise.
Authors' answer
Corrected, thanks
See L507-510
“Here, we revealed that KA exposure caused RhoA protein levels to significantly increase 48 h after exposure compared to control conditions (p⩽0.01, n= 6 independent WB experiments including biological triplicates and 2 technical replicates, Figure 6A-B).”
- Reviewer comments
Paragraph 4.7 – “We therefore investigated the efficiency of exogenous BDNF and NT3 to boost synapse regeneration.” Using the term “synapse regeneration” requires additional experiments, which were not performed. The authors studied changes in the number of synapses, which could result from synaptic regeneration, de novo synaptogenesis, inhibition of synaptopathy, or all of the above.
Authors' answer
The reviewer is right; we have therefore rephrased this sentence.
See L541-542
“We therefore investigated the efficacy of exogenous BDNF and NT3 to rescue the IHC synapses from KA-induced excitotoxicity.”
- Reviewer comments
Figure 5A – the images are illegible. Please consider using larger micrographs.
Authors' answer
Done, thanks
See new figure 7
- Reviewer comments
Lines 560-561 – “Together, these results strongly support our hypothesis that glutamatergic excitotoxicity-induced loss of synapses is mediated by increased production of ROS and oxidative stress” – I suggest communicating this more cautiously, e.g., “(…) is mediated at least in part by increased production….”.
Authors' answer
Performed, thanks!
See L655-658
“Together, these results strongly support our hypothesis that glutamatergic excitotoxicity-induced loss of synapses is mediated, at least in part, by increased production of ROS and oxidative stress and that cochlear synaptopathy could be mitigated by antioxidant treatments.”
- Reviewer comments
The syntax and grammar need polishing.
Authors' answer
The manuscript has been edited by a native English speaker to correct any language errors.
